# Effects of Phytonutrients on Ruminal Fermentation, Digestibility, and Microorganisms in Swamp Buffaloes

**DOI:** 10.3390/ani9090671

**Published:** 2019-09-11

**Authors:** Thiwakorn Ampapon, Kampanat Phesatcha, Metha Wanapat

**Affiliations:** 1Tropical Feed Resources Research and Development Center (TROFREC), Department of Animal Science, Faculty of Agriculture, Khon Kaen University, Khon Kaen 40002, Thailand; 2Department of Animal Science, Faculty of Agriculture and Technology, Nakhon Phanom University, Nakhon Phanom 48000, Thailand

**Keywords:** rumen enhancer, phytonutrients, fruit waste, greenhouse gas

## Abstract

**Simple Summary:**

Fermentation in ruminants can influence greenhouse gas production, especially methane (CH_4_) production. Phytonutrients (PTN) and secondary metabolites (tannins, saponins) have antimicrobial activity which can be used for the inhibition of rumen methanogens and the reduction of methane emissions in ruminants. The aim of the present study was to investigate the influence of PTN containing both mangosteen peel powder and banana flower powder on feed consumption, nutrient digestibility, and rumen microorganisms in swamp buffaloes. The results suggest that supplementation of PTN can improve fiber digestibility, increase the proteolytic and cellulolytic bacteria, and alter rumen volatile fatty acids (VFAs), especially increased C_3_ and reduced methane production. This study shows that PTN containing either mangosteen peel powder (MSP) or banana flower powder (BFP) could be used as a rumen modifier.

**Abstract:**

This experiment aimed to use dietary sources containing phytonutrients (PTN) such as mangosteen peel powder (MSP) and banana flower powder (BFP) as sources of phytonutrients. Four swamp buffalo bulls fitted with rumen fistulae were used as experimental animals. A digestion trial covering four periods was used according to a 4 × 4 Latin square design with four treatments: Treatment 1 (T1) = control (Cont), T2 = supplementation of PTN1 fed at 100 g/d, T3 = supplementation of PTN2 fed at 100 g/d, and T4 = supplementation of PTN3 fed at 100 g/d. The experiment was conducted for four periods; each period lasted for 21 days. All animals were fed a concentrate mixture at 0.5% body weight, while rice straw, water, and mineral blocks were fed ad libitum. The findings revealed significant increases in the digestibility of neutral detergent fiber (NDF) and acid detergent fiber (ADF), while no changes in dry matter feed consumption occurred due to PTN supplementation. Rumen fermentation end-products, such as total volatile fatty acids (TVFA), propionic acid (C_3_), and butyric acid (C_4_), were notably enhanced (*p* < 0.05) and there were the highest in PTN2 and PTN3, whilst acetic acid (C_2_) was significantly decreased with PTN supplementation groups. Furthermore, the rumen protozoal population was suppressed (*p* < 0.05), which resulted in decreased rumen methane production (*p* < 0.05), while the bacterial population was enhanced. Using PTN sources can improve rumen fermentation as well as mitigating rumen methane production.

## 1. Introduction

There are many factors that could alter rumen ecology, for example, a high grain and low fiber diet can reduce rumen pH. Feed additives like antibiotics or chemicals and buffers can improve rumen fermentation efficiency [1]. Research and development regarding methane (CH_4_) production in ruminants has received considerable attention in which mitigation of the rumen CH_4_ has been the main issue [2,3]. CH_4_ production from rumen fermentation can result in a loss of energy intake of about 8–12% in ruminants [4]. Phytonutrients or secondary metabolites (tannins, saponins, essential oils, etc.) impose antimicrobial activity which can be used for the inhibition of specific microorganism groups in the rumen and can modify their fermentation [5,6]. Poungchompu et al. [7] and Shokryzadan et al. [8] stated that mangosteen (*Garcinia mangostana*) peel is one kind of agricultural by-product containing the highest levels of condensed tannins (CT) and saponins (SP) when compared with other tropical plants and fruit peels. It is a non-chemical feed additive that is used as a rumen fermentation modifier, and it mitigates methane production. Phytonutrients have been shown to affect ruminal fermentation efficiency, especially by decreasing the acetic acid (C_2_) concentration, reducing ruminal protozoa and the methanogen population, and mitigating CH_4_ emissions [9]. Furthermore, Wanapat et al. [10] reported that consumption of dietary mangosteen peel powder (MSP) by buffaloes can increase the concentration of C_3_ and microbial protein synthesis. In addition, Oskoueian et al. [11] found that flavonoids, especially naringin and quercetin, suppressed the protozoal population and methanogen population in an in vitro experiment. 

Banana (*Musa sapientum* L.) flower powder (BFP) has been reported to be used as a rumen buffering agent, as it contains a high concentration of mineral elements. The pH stability in the rumen is considered a prime factor that feeds into optimal rumen ecology, as the shift towards acidic conditions, especially pH values lower than 6.0, will dramatically affect microorganisms and the fermentation process [1,12]. Dietary sources containing phytonutrients (PTN) have been used to replace chemical buffers and have resulted in improved rumen fermentation, as shown by Poungchonpu et al. [7], Norrapoke et al. [13], Foiklang et al. [14], and Kang et al. [15,16]. However, the use of other PTN in combination, especially in swamp buffaloes, has not been fully investigated. Therefore, this experiment aimed to investigate the influence of PTN containing both mangosteen peel powder and banana flower powder on feed consumption, nutrient digestibility, and rumen microorganisms in swamp buffaloes fed rice straw as a basal roughage. 

## 2. Materials and Methods

### 2.1. Feeds, Animals, and Experimental Design

The sources of phytonutrients were as follows: The mangosteen peel (MSP) was collected from fresh fruit, sun-dried for 3 to 4 days, and ground into powder form. The banana flower powder (BFP) was collected from fresh flowers, sun-dried for 5 to 6 days, and ground into powder form. Mangosteen peel powder, banana flower powder, urea, and sulfur were mixed to form PTN1, PTN2, and PTN3, as reported in Table 1. The concentrate contained 14.5% of crude protein. Feed ingredients and chemical compositions are presented in Table 1.

The study was conducted at the Tropical Feed Resources Research and Development Center (TROFREC), Department of Animal Science, Faculty of Agriculture, Khon Kaen University (KKU), Thailand. All procedures involving animals in the metabolism studies were approved by the Institutional Animal Care and Use Committee of Khon Kaen University (KKU) (ref. no. AEKKU 18/2558).

Four swamp buffalo bulls fitted with rumen fistulae were used as experimental animals. A digestion trial covering four periods was used in a 4 × 4 Latin square design with four treatments: Treatmeat1 (T1) = control (Cont), T2 = supplementation of PTN1 fed at 100 g/d, T3 = supplementation of PTN2 fed at 100 g/d, and T4 = supplementation of PTN3 fed at 100 g/d. The experiment was conducted for 4 periods, and each period lasted for 21 days. All animals were fed the concentrate mixture at 0.5% body weight, while the concentrate mixture was fed in the morning and afternoon (07:00 a.m., 16:30 p.m.). Rice straw was fed ad libitum. Water and mineral blocks were available at all times for all buffaloes.

### 2.2. Data Collection and Sample Analysis

After the first 14 days, buffaloes had adapted to the diet, and feed consumption was measured. Samples of concentrate and rice straw including refusals were collected daily during the feeding period. During the last 7 days, all buffaloes were moved to the metabolism crates for feed, feces, and urine collection. Feed and fecal samples were measured for nutritive values of dry matter (DM), organic matter (OM), ash and crude protein (CP) [17] and neutral detergent fiber (NDF), acid detergent fiber (ADF), and acid detergent lignin (ADL) [18], and then nutrient digestibility was estimated according to the standard methods described by Wanapat et al. [19]. Condensed tannins (CT) were analyzed by the Vanillin-HCL method (Burns [20]; modified by Wanapat and Poungchompu [21]).

At the last day of each period, rumen fluid samples were collected at 0, 2, 4, and 6 h post morning feeding. Rumen fluid was immediately measured for pH and temperature using a portable pH meter (HANNA Instrument HI 8424 microcomputer, Hanna Instruments (S) Pte Ltd., 161 Kallang Way, Singapore). Rumen fluid samples were then filtered through three layers of cheesecloth. The first part was used for the analysis of volatile fatty acids (VFA) and NH_3_–N, where 5 mL of H_2_SO_4_ solution (1 M) was added to 45 mL of rumen fluid. The mixture was centrifuged at 1600× *g* for 15 min, and the supernatant was stored at −20 °C prior to VFA analysis using high-performance liquid chromatography (HPLC) and the NH_3_–N analysis method [17]. The next part was used for the measurement of microbial populations by total direct counts of bacteria, protozoa, and fungal zoospores [22]. The last part was cultured for groups of bacteria (i.e., cellulolytic, proteolytic, amylolytic, and total viable bacterial counts) using the roll-tube technique [23]. A blood sample (about 10 mL) was drawn from the jugular vein into EDTA-containing tubes and was separated by centrifugation at 500× *g* for 10 min at 4 °C to obtain plasma. Then, the plasma was stored at −20 °C until the analysis of blood urea N according to the method of Crocker [24]. 

Rumen CH_4_ production was estimated using VFA proportions according to Moss et al. [25] and as follows: CH_4_ production = 0.45(acetate, C_2_) − 0.275(propionate, C_3_) + 0.4(butyrate, C_4_).

### 2.3. Statistical Analyses

All data were statistically analyzed using the general linear-model procedure (Statistical Analysis System, SAS (2013) [26]) according a 4 × 4 Latin square design in which the buffalo, period, and level of PTN supplementation were the main factors. Treatment means were analyzed using the following model: Yijk = μ + Ti + Cj + Rk + eijk, where Yijk is the criteria under study in treatment i, column j, and row k; μ is the overall sample mean; Ti is the effect of treatment i; Cj is the effect of treatment i at column j; Rk is the effect of treatment i at row k; and eijk is the error.

## 3. Results

### 3.1. Voluntary Feed Consumption and Digestibility

In Table 2, rice straw intake was not influenced by PTN supplementation, and total DM feed intake was not significantly different among PTN supplementation groups. The apparent digestibility in terms of DM, OM and CP was not increased by PTN supplementation (*p* > 0.05). However, the PTN groups showed significantly enhanced digestibility of fiber (NDF, ADF). Although the values were the highest for the PTN2 and PTN3 groups, there were no significant difference among treatments.

### 3.2. Ruminal Fermentation, Blood, Methane Production

The results for ruminal fermentation, blood, and methane production are presented in Table 3. The pH (6.6–6.8) and temperature (38.8–39.5) in the rumen were unchanged (*p* > 0.05). The ruminal NH_3_–N concentration was increased in the PTN supplementation groups, while the blood urea nitrogen (BUN) concentration was similar among treatments. The total volatile fatty acid (TVFA), propionic acid (C_3_), and butyric acid (C_4_) concentrations increased (*p* < 0.05) following PTN supplementation, especially in the PTN2 and PTN3 groups, which had the highest concentrations. In contrast, the acetic acid (C_2_) concentration decreased (*p* < 0.05), while the C_2_:C_3_ ratio and estimated CH_4_ production decreased remarkably in the PTN supplementation groups, and was the lowest in PTN2. 

### 3.3. Microbial Population Enumeration

The protozoal population significantly decreased (*p* < 0.05) following PTN supplementation, while under PTN1 and PTN3 supplementations, the protozoal numbers remarkably decreased (*p* < 0.05). However, the concentrations of fungal zoospores, starch-digesting bacteria, and total bacteria were unchanged (*p* > 0.05) among treatments, while the rumen proteolytic and cellulolytic bacterial populations were the highest (*p* < 0.05) in the PTN2 and PTN3 supplementation groups (Table 4). 

## 4. Discussion

### 4.1. Voluntary Feed Consumption and Digestibility

Supplementation of PTN did not affect feed consumption in this experiment. This is similar to the findings of Kang and Wanapat [27], who reported that supplementation of a banana flower powder (BFP)-pellet at 4% of dry matter intake (DMI) did not impact feed intake in dairy steers. The use of MSP at 30 g/d did not impact feed consumption and nutrient digestibility in buffaloes [28], while the use of tannin sources as less than 50 g/kg DM of feed did not suppress DMI or nutrient digestibility in beef cattle [29]. However, Cruywagen et al. [30] found that the use of bicarbonate as a buffering agent in a high concentrate diet did not alter dry matter feed intake. Moreover, the result from Rauch et al. [31] revealed that the use of sodium bicarbonate and calcium magnesium carbonate did not influence dry matter feed consumption but increased the digestibility of crude protein in lactating dairy cows. In this study, DM, OM and CP digestibility did not ultimately change, while NDF and ADF contrarily increased in the PTN2 and PTN3 groups. This phenomenon may be because BFP and CT reduce the protozoal number, and bacteria are the feed substrate of protozoa; hence, it would be expected that the fibrolytic bacteria population would subsequently increase. In addition, BFP supplementation improves the in vitro true digestibility, according to Kang and Wanapat [27]. Furthermore, rumen pH is the major factor influencing fibrolytic bacteria attachment in the particles of feed [32]. 

### 4.2. Ruminal Fermentation, Blood, and Methane Production

In this experiment, the pH and temperature in the rumen were not impacted and remained in the narrow range for ruminal fermentation of feeds [19,32]. Russell [33] stated the importance of pH on propionate production in the rumen; hence, the pH should be maintained for optimal rumen ecology. The rumen NH_3_–N concentration increased in the PTN supplementation groups, while BUN was unchanged. The optimal level of NH_3_–N is from 15 to 30 mg/dL for rumen fermentation, as revealed by Preston and Leng [34] and Wanapat and Pimpa [35] under a rice straw feeding regime. The TVFA, C_3_, and C_4_ were significantly increased, especially in the PTN2 and PTN3 groups, and were the highest, while C_2_ and the C_2_:C_3_ ratio decreased following PTN supplementation. The reason for this is probably that the buffering agent and CT from PTN sources resulted in enhanced fiber digestion and microbial activity. Similarly, Cruywagen et al. [30] reported that the use of buffering agent from sodium bicarbonate and limestone in the high concentrate diet could increase the total VFA and reduce lactate fermentation. Moreover, CH_4_ production was consequently decreased by PTN supplementation. This could be the result of the suppression of protozoa and methanogens in the rumen. Similarly, it was found that the use of mangosteen peel pellets and soapberry fruit supplementation containing phytonutrients can influence ruminal fermentation by decreasing the C_2_ concentration and mitigating CH_4_ production, thereby increasing the C_3_ concentration [7]. Wanapat et al. [10] reported that the use of MSP at 100 g/d in swamp buffaloes affected the total VFA concentration and increased C_3_, reducing C_2_:C_3_ and methane production following MSP supplementation. Recently, Shokryzaden et al. [8] and Paengkoum et al. [36] found that condensed tannins (CT) from MSP could reduce methane production by the suppression of ruminal microbes, especially protozoa and methanogens. 

### 4.3. Rumen Microorganism Population 

The protozoal population was suppressed by supplementation, especially in the PTN1 and PTN3 supplementation groups. This could be due to CT in PTN, which directly react with sterols in the membranes of protozoa. This result was confirmed by Ngamseang et al. [37], who found that the use of MSP containing CT at 150 g/d suppressed protozoal population, while supplementation with MSP and soapberry fruit pellets containing CT and SP suppressed protozoal population [7]. Similarly, Shokryazdan et al. [8] additionally stated that replacing alfalfa with 25% and 50% MSP mitigated the protozoal population. Polyorach et al. [38] reported, in addition, that MSP supplementation in cows increased the total bacteria concentration, while the protozoal and methanogen populations reduced. Suppression of the rumen protozoal population was found in dairy steers and dairy cows as a result of MSP supplementation [13,37]. Furthermore, Dong et al. [39] revealed that the dietary supplementation of *Moringa oleifera* containing plant phytonutrients changed the composition and diversity of methanogens and reduced methane emissions in dairy cows. In this study, the fungal zoospore, amylolytic bacteria, and total bacteria concentrations did not differ among treatment groups. However, the cellulolytic bacteria and proteolytic bacteria concentrations increased in the PTN2 and PTN3 supplementation groups. This may be due to the enhancement of the rumen pH by BFP supplementation [16]. This result was confirmed by Kang and Wanapat [27], who used BFP as a rumen pH buffer, and the results revealed an increase in the rumen pH, as BFP constitutes a high level of minerals, especially Ca, Mg, and K, and the use of BFP could increase nutrient digestibility and improve rumen fermentation in the ruminants.

## 5. Conclusions

Some source of phytonutrients (PTN) can significantly improve fiber digestibility, bacterial population (proteolytic and cellulolytic bacteria) concentrations, and rumen VFA, especially through an enhanced C_3_ concentration and, consequently, reduced CH_4_ production. This study suggests that PTN3 containing either mangosteen peel powder (MSP) or banana flower powder (BFP) could be recommended as a rumen modifier to enhance the rumen fermentation efficiency without negatively affecting feed consumption or the rumen fermentation process.

## Figures and Tables

**Table 1 animals-09-00671-t001:** Feed ingredients and chemical compositions of the experimental diets.

Item	Concentrate	PTN1	PTN2	PTN3	Rice Straw
Ingredients, g/kg dry matter					
Cassava chip	618	-	-	-	-
Rice bran	78	-	-	-	-
Coconut meal	120	-	-	-	-
Palm kernel meal	120	-	-	-	-
Mangosteen peel powder (MSP)	-	910	-	455	-
Banana flower powder (BFP)	-	-	910	455	-
Cassava chip meal	-	50	50	50	-
Molasses	19	-	-	-	-
Urea	30	30	30	30	-
Mineral mixture	5	-	-	-	-
Salt	5	-	-	-	-
Sulfur	5	10	10	10	-
Chemical composition, g/kg dry matter					
Dry matter, g/kg	898	925	905	913	905
Organic matter	920	942	913	921	880
Ash	80	58	87	79	220
Crude protein	145	189	151	168	29
Neutral detergent fiber	265	557	677	611	795
Acid detergent fiber	124	495	451	475	436
Condensed tannins	-	167	113	145	-

PTN1 (containing mangosteen peel powder at 91%) at 100 g/head/day; PTN2 (containing banana flower powder at 91%); PTN3 (containing mangosteen peel powder at 45.5% and banana flower powder at 45.5%).

**Table 2 animals-09-00671-t002:** Effects of some sources of phytonutrients (PTN) on voluntary feed intake and nutrient digestibility in swamp buffaloes.

Items	Cont	PTN1	PTN2	PTN3	SEM	*p*-Value
Rice straw DM intake						
kg/day	6.8	6.7	6.9	6.9	0.20	0.87
%BW	1.4	1.4	1.4	1.4	0.04	0.40
Concentrate DM intake						
kg/day	1.8	1.9	1.9	1.9	0.32	0.19
%BW	0.5	0.5	0.5	0.5	0.07	0.61
Total DM intake						
kg/day	8.6	8.8	8.8	8.9	0.13	0.56
%BW	1.9	1.9	1.9	1.9	0.25	0.47
Apparent digestibility, %DM						
Dry matter, %	59.0	58.5	57.9	58.7	1.56	0.45
Organic matter	61.2	60.3	60.1	61.0	1.77	0.33
Crude protein	62.5	67.2	64.8	65.7	2.45	0.72
Neutral detergent fiber	40.5 ^a^	45.7 ^b^	48.1 ^b^	48.0 ^b^	1.15	0.01
Acid detergent fiber	40.2 ^a^	45.4 ^b^	49.7 ^c^	48.5 ^c^	0.45	0.01

^a,b,c^ Means in the same row with different superscripts differ (*p* < 0.05); Cont (control group); PTN1 (containing mangosteen peel powder at 91%) at 100 g/head/day; PTN2 (containing banana flower powder at 91%); PTN3 (containing mangosteen peel powder at 45.5% and banana flower powder at 45.5%); SEM = standard error of the mean.

**Table 3 animals-09-00671-t003:** Effects of some sources of phytonutrients (PTN) on the ruminal pH, temperature, ammonia nitrogen (NH_3_–N), blood urea nitrogen (BUN), and volatile fatty acid concentrations in swamp buffaloes.

Item	Supplementation (100 g/h/day)				SEM	*p*-Value
	Cont	PTN1	PTN2	PTN3		
Rumen parameters						
pH	6.7	6.6	6.8	6.8	1.89	0.13
Temperature	38.8	38.9	39.5	39.1	1.53	0.35
Ammonia nitrogen, mg/dL	12.1 ^a^	15.7 ^b^	14.0 ^a^	14.9 ^b^	0.60	0.01
Blood urea nitrogen, mg/dL	6.8	6.6	6.6	6.7	2.77	0.24
Total volatile fatty acid, mmol/L	97.2 ^a^	101.0 ^ab^	103.0 ^b^	106.6 ^b^	1.71	0.04
Acetic acid (C_2_), %	73.4 ^a^	65.1 ^b^	59.3 ^c^	64.0 ^b^	1.12	0.01
Propionic acid (C_3_), %	15.5 ^a^	21.5 ^b^	26.9 ^c^	22.5 ^b^	1.15	0.01
Butyric acid (C_4_), %	11.1 ^a^	13.0 ^b^	14.9 ^c^	13.6 ^b^	0.41	0.02
C_2_:C_3_	4.8 ^a^	3.2 ^b^	2.1 ^c^	2.9 ^b^	0.57	0.01
Methane estimation, mM/L *	33.2 ^a^	28.7 ^b^	25.8 ^c^	28.1 ^b^	0.83	0.03

^a,b,c^ Means in the same row with different superscripts differ (*p* < 0.05); Cont (control group); PTN1 (containing mangosteen peel powder at 91%) at 100 g/head/day; PTN2 (containing banana flower powder at 91%); PTN3 (containing mangosteen peel powder at 45.5% and banana flower powder at 45.5%); * Calculated according to Moss et al. [25] CH_4_ production = 0.45(acetate) − 0.275(propionate) + 0.4(butyrate); SEM = standard error of the mean.

**Table 4 animals-09-00671-t004:** Effects of some sources of phytonutrients (PTN) on the rumen protozoa, fungal zoospore, and bacteria populations in swamp buffaloes.

Items	Supplementation (100 g/h/day)				SEM	*p*-Value
	Cont	PTN1	PTN2	PTN3		
Total direct counts						
Protozoa, ×10^5^ cell/mL	8.2 ^a^	6.5 ^b^	7.7 ^c^	6.7 ^b^	0.09	0.02
Fungal zoospore, ×10^5^ cell/mL	5.4	5.4	5.6	5.6	0.41	0.12
Roll-tube technique, CFU/mL Amylolytic bacteria, ×10^7^	2.9	3.2	3.3	3.2	0.43	0.07
Proteolytic bacteria, ×10^7^	3.5 ^a^	4.2 ^b^	4.4 ^b^	4.4 ^b^	0.15	0.02
Cellulolytic bacteria, ×10^8^	3.4 ^a^	3.5 ^a^	4.6 ^b^	4.1 ^b^	0.16	0.01
Total viable bacteria, ×10^8^	5.4	5.6	5.9	5.7	0.77	0.15

^a,b,c^ Means in the same row with different superscripts differ (*p* < 0.05); Cont (control group); PTN1 (containing mangosteen peel powder at 91%) at 100 g/head/day; PTN2 (containing banana flower powder at 91%); PTN3 (containing mangosteen peel powder at 45.5% and banana flower powder at 45.5%); SEM = standard error of the mean.

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
