# Peer review of "Effects of Phytonutrients on Ruminal Fermentation, Digestibility, and Microorganisms in Swamp Buffaloes"

_animals, 2019, doi:10.3390/ani9090671_

Round 1

Reviewer 1 Report

The paper animals-527568 entitled “Dietary rumen enhancer (DRE) enhancing on ruminal fermentation, digestibility and microorganisms in swamp buffaloes” deals with an interesting topic on using of dietary rumen enhancer based on phytonutrients such as mangosteen peel powder and banana flower powder and their combination.  The topic is of recently interest, particularly in the context of circular economy. The research was well conducted for the methodologies and experimental design section; however, some critical suggestions emerged as regard the writing style and results interpretation. 

Following the paper construction, the introduction section needs to include a specific part and some references on ruminant study and in particular on previous researches. 

Comments at lines:

L46: Please substitute “their fermentation” with “modifying their fermentation”. 

L48: The mangosteen peel contains the highest condensed tannins in compared with??Please add more information. 

L63: Please delete point after 16]). 

L64-66: In the objective of the study is missed the referment on combination of MSP and BFP. In my opinion and by scientific evidence on the use of tannins as dietary supplement for different purpose, the effects of tannins can be boosted by the combination of different form of tannins (hydrolyzed and condensed) or, such as hypothesized in this paper by the combination of two by-products. The overall result of the tannins combination could be the synergistic action, mainly when have different composition as emerged by chemical composition of experimental diet. Please add some of these observation into introduction section. 

L69: Please describe the diet preparation without using 1, 2, and 3. 

Did the authors performed any tannins characterization present in MSP and BFP DRE? If yes, I suggest to report them in the table 1.

Table 1: Please control the organic matter value for concentrate, maybe 92.0 is wrong. 

L94: Please adjust “After the first 14 days, buffaloes were adapted to the diet and feed consumption parameters were recorded. Sample…”. 

L104: Read L69. 

L112: “to obtain plasma stored at -20°C until analysis”. 

Results section

L130: It is not true, because the P value is P=0.01 for ADF and NDF. 

L141: P value are respectively P=0.04 for TVFA, P=0.01 for C3and P=0.02 for C4

L143: The results regard the methane production are missing. However, in the discussion section at L196-197 some results have been presented. 

L180: Please substitute “important” with “importance”.

Discussion section 

The first paragraph is concentrated on buffering agent discussion and its relation with dry matter intake, no mention on MSP effect was added. Please specify at the light of the previous comment; moreover, the combination of MSP and BFP needs to be discussed.

L199-202: please rewrite as not a result. 

Conclusion section

The main result is on DRE3 that represent the combination of two treatment; however, no synergistic action was supposed into discussion and conclusion section. 

Author Response

Dear Editor,

Thank you for your letter and for the reviewers’ comments concerning our manuscript entitled “Effect of using some sources of phytonutrients on ruminal fermentation, digestibility and microorganisms in swamp buffaloes” (Manuscript ID: animals-527568). Those comments are all valuable and very helpful for revising and improving our paper, as well as the important guiding significance to our researches. We have studied comments carefully and have made corrections which we hope to you’re your approval. Revised portion are marked in yellow in the paper.

We appreciate for Editors/Reviewers’ constructive work earnestly, and hope that the corrections will meet with approval.

Once again, thank you very much for your comments and suggestions.

We look forward to your prompt reply.

Sincerely yours,

Prof. Dr. Metha Wanapat

On behalf of authors

Contact: metha@kku.ac.th

The main corrections in the paper and the responds to the reviewer’s comments are as flowing:

Reviewer 1

Comments and Suggestions for Authors

The paper animals-527568 entitled “Dietary rumen enhancer (DRE) enhancing on ruminal fermentation, digestibility and microorganisms in swamp buffaloes” deals with an interesting topic on using of dietary rumen enhancer based on phytonutrients such as mangosteen peel powder and banana flower powder and their combination.  The topic is of recently interest, particularly in the context of circular economy. The research was well conducted for the methodologies and experimental design section; however, some critical suggestions emerged as regard the writing style and results interpretation.

Following the paper construction, the introduction section needs to include a specific part and some references on ruminant study and in particular on previous researches.

Comments at lines:

L46: Please substitute “their fermentation” with “modifying their fermentation”.

Response: Thanks, we have revised already.

L48: The mangosteen peel contains the highest condensed tannins in compared with??Please add more information.

Response: Thanks, we have added already.

L63: Please delete point after 16]).

Response: Thanks, we have revised already.

L64-66: In the objective of the study is missed the referment on combination of MSP and BFP. In my opinion and by scientific evidence on the use of tannins as dietary supplement for different purpose, the effects of tannins can be boosted by the combination of different form of tannins (hydrolyzed and condensed) or, such as hypothesized in this paper by the combination of two by-products. The overall result of the tannins combination could be the synergistic action, mainly when have different composition as emerged by chemical composition of experimental diet. Please add some of these observation into introduction section.

Response: Thanks, we have revised already.

L69: Please describe the diet preparation without using 1, 2, and 3.

Response: Thanks, we have revised already.

Did the authors performed any tannins characterization present in MSP and BFP DRE? If yes, I suggest to report them in the table 1.

Response: Thanks, we have added already.

Table 1: Please control the organic matter value for concentrate, maybe 92.0 is wrong.

L94: Please adjust “After the first 14 days, buffaloes were adapted to the diet and feed consumption parameters were recorded. Sample…”.

Response: Thanks, we have revised already.

L104: Read L69.

Response: Thanks, we have revised already.

L112: “to obtain plasma stored at -20°C until analysis”.

Response: Thanks, we have revised already.

Results section

L130: It is not true, because the P value is P=0.01 for ADF and NDF.

Response: Thanks, we have revised already.

L141: P value are respectively P=0.04 for TVFA, P=0.01 for C3and P=0.02 for C4

Response: Thanks, we have revised already.

L143: The results regard the methane production are missing. However, in the discussion section at L196-197 some results have been presented.

Response: Thanks, we have revised already.

L180: Please substitute “important” with “importance”.

Response: Thanks, we have revised already.

Discussion section

The first paragraph is concentrated on buffering agent discussion and its relation with dry matter intake, no mention on MSP effect was added. Please specify at the light of the previous comment; moreover, the combination of MSP and BFP needs to be discussed.

Response: Thanks, we have added and revised already.

L199-202: please rewrite as not a result.

Response: Thanks, we have revised already.

Conclusion section

The main result is on DRE3 that represent the combination of two treatment; however, no synergistic action was supposed into discussion and conclusion section.

Response: Thanks, we have added and revised already.

Reviewer 2 Report

1.      It is better to change the title to: "Effect of using some sources of phytonutrients on ruminal fermentation, digestibility and microorganisms in swamp buffaloes".

2.      Use assistance of a professional English editor to revise the language of this study.

3.      In line 22 replace "dietary rumen enhances" with "sources of phytonutrients".

4.      It is best to write the scientific names to the mangosteen and banana.

5.      It is better to standardize abbreviations for treatments in all the paper as follows:  T1 = control, T2 = (MSP)  ,T3 = (BFP) and T4 = (MSP+BFP).

6.      The duration of the experiment should be stated and the number of animals clarified in abstract.

7.      Estimation of methane production based on arithmetical equations   is not accurate (line 116).

8.      Table 1 is not understood, for example DRE1 consists of Mangosteen peel powder and Cassava chip meal only!!!

9.      Supplementation of (MSP) and (BFP) fed at 100 g/d or 910 g/kg DM (table1)… Not understood.

10.  How and when the (MSP) and (BFP)   are added to animals?

11.  Crude protein significantly increased in the animal's rations for T2, T3andT4 (Table 1), this may affect the results of the experiment, it was supposed to re-adjust the animal rations so as not to be an effective factor.

12.  Explain the abbreviation of %BW in Table 2.

13.  The values of methane production are eliminated from the results section and are not discussed because they are not estimated by chemical methods.

14.  Most of the discussion is a review of previous research, I hope to focus more in your results and explain why the differences are significant in some results.

Author Response

Dear Editor,

Thank you for your letter and for the reviewers’ comments concerning our manuscript entitled “Effect of using some sources of phytonutrients on ruminal fermentation, digestibility and microorganisms in swamp buffaloes” (Manuscript ID: animals-527568). Those comments are all valuable and very helpful for revising and improving our paper, as well as the important guiding significance to our researches. We have studied comments carefully and have made corrections which we hope to you’re your approval. Revised portion are marked in yellow in the paper.

We appreciate for Editors/Reviewers’ constructive work earnestly, and hope that the corrections will meet with approval.

Once again, thank you very much for your comments and suggestions.

We look forward to your prompt reply.

Sincerely yours,

Prof. Dr. Metha Wanapat

On behalf of authors

Contact: metha@kku.ac.th

Reviewer 2

1.      It is better to change the title to: "Effect of using some sources of phytonutrients on ruminal fermentation, digestibility and microorganisms in swamp buffaloes".

Response: Thanks, we have revised already.

2.      Use assistance of a professional English editor to revise the language of this study.

Response: Thanks, we have carefully and have made corrections.

3.      In line 22 replace "dietary rumen enhances" with "sources of phytonutrients".

Response: Thanks, we have revised already.

4.      It is best to write the scientific names to the mangosteen and banana.

Response: Thanks, we have revised already.

5.      It is better to standardize abbreviations for treatments in all the paper as follows:  T1 = control, T2 = (MSP)  ,T3 = (BFP) and T4 = (MSP+BFP).

Response: Thanks, we have revised already.

6.      The duration of the experiment should be stated and the number of animals clarified in abstract.

Response: Thanks, we have revised already.

7.      Estimation of methane production based on arithmetical equations   is not accurate (line 116).

Response: We have change to prediction of rumen methane production was done by using VFA proportions in the equation of Moss et al. (2000).

8.      Table 1 is not understood, for example DRE1 consists of Mangosteen peel powder and Cassava chip meal only!!!

Response: Thanks, we have revised. Please see detail in the table.

9.      Supplementation of (MSP) and (BFP) fed at 100 g/d or 910 g/kg DM (table1)… Not understood.

Response: Thanks, we have revised in material method already, as shown in the text.

10.  How and when the (MSP) and (BFP)   are added to animals?

Response: the mangosteen peel (MSP), banana flower powder (BFP), urea and sulfur were mixed for PTN1, PTN2 and PTN3 and then fed to animals at 100 g/hd/day.

11.  Crude protein significantly increased in the animal's rations for T2, T3andT4 (Table 1), this may affect the results of the experiment, it was supposed to re-adjust the animal rations so as not to be an effective factor.

Response: Thanks, we have revised and rechecked data already, as shown in the text.

12.  Explain the abbreviation of %BW in Table 2.

Response: %BW of dry mater intake. We can calculate by the quantity intake and the weight of animals.

13.  The values of methane production are eliminated from the results section and are not discussed because they are not estimated by chemical methods.

Response: We wish to contain discussion of rumen methane production, since it was based on prediction of fermentation end-products especially those of acetate, propionate and butyrate.

14.  Most of the discussion is a review of previous research, I hope to focus more in your results and explain why the differences are significant in some results.

Response: Thanks, we have revised already.

Round 2

Reviewer 1 Report

The paper has been ameliorated in all parts in the revised version; some minor comments have been added:

L 87 Please add Cont without point to indicate control group as reported in the table 2 in which the authors can delete the point after Cont

L 130 Change “there were no difference among treatments”. 

L138 Add a brief introduction such as “The results on ruminal fermentation, blood and methane production are presented in Table 3”. 

L 169 Please substitute “whilst” that has been already used at line 168. 

L190: Please use only CT instead of “condensed tannins (CT). 

Author Response

Dear Editor,

Thank you for your letter and for the reviewers’ comments concerning our manuscript entitled “Effect of using some sources of phytonutrients on ruminal fermentation, digestibility and microorganisms in swamp buffaloes” (Manuscript ID: animals-527568). Those comments are all valuable and very helpful for revising and improving our paper, as well as the important guiding significance to our researches. We have studied comments carefully and have made corrections which we hope to you’re your approval. Revised portion are marked in yellow in the paper.

We appreciate for Editors/Reviewers’ constructive work earnestly, and hope that the corrections will meet with approval.

Once again, thank you very much for your comments and suggestions.

We look forward to your prompt reply.

Sincerely yours,

Prof. Dr. Metha Wanapat

On behalf of authors

Contact: metha@kku.ac.th

The main corrections in the paper and the responds to the reviewer’s comments are as flowing:

Reviewer 1

Comments and Suggestions for Authors

The paper has been ameliorated in all parts in the revised version; some minor comments have been added:

L 87 Please add Cont without point to indicate control group as reported in the table 2 in which the authors can delete the point after Cont

Response: Thanks, we have already shown all details in the text. 

L 130 Change “there were no difference among treatments”.

Response: Thanks, we have revised already.

L138 Add a brief introduction such as “The results on ruminal fermentation, blood and methane production are presented in Table 3”.

Response: Thanks, we have already shown all details in the text.

L 169 Please substitute “whilst” that has been already used at line 168.

Response: Thanks, we have revised already.

L190: Please use only CT instead of “condensed tannins (CT).

Response: Thanks, we have revised already.

Reviewer 2

Comments and Suggestions for Authors

The authors did not clarify these queries:

1.     Table 1 is not understood, for example, DRE1 consists of Mangosteen peel powder and Cassava chip meal only!!!

Response: We have already shown all details in the text.  

- PTN1 containing MSP = 91%, cassava chip meal = 5%, urea = 3% and sulfur 1%.

- PTN2 containing BFP = 91%, cassava chip meal = 5%, urea = 3% and sulfur 1%.

- PTN3 containing MSP = 45.5%, BFP = 45.5%, cassava chip meal = 5%, urea = 3% and sulfur 1%.

2.     Supplementation of (MSP) and (BFP) fed at 100 g/d or 910 g/kg DM (table1)… Not understood.

Response: The three sources of PTN were MSP (PTN1), BFP (PTN2) and MSP+BFP (PTN3) were offered to the animals daily at 100 g/head/day. As shown in the text.        

3.     How and when the (MSP) and (BFP)   are added to animals?

Response: The supplements of PTN1, PTN2 and PTN3 were given to the animals by mixing in the concentrate mixture at each feeding time.  

4.     Explain the abbreviation of %BW in Table 2.

Response: %BW = DMI were calculated as amount eaten as a percentage of animal liveweight.  

Reviewer 2 Report

The authors did not clarify these queries:

1.     Table 1 is not understood, for example, DRE1 consists of Mangosteen peel powder and Cassava chip meal only!!!

2.     Supplementation of (MSP) and (BFP) fed at 100 g/d or 910 g/kg DM (table1)… Not understood.

3.     How and when the (MSP) and (BFP)   are added to animals?

4.     Explain the abbreviation of %BW in Table 2.

Author Response

(The authors gave the same response as above.)
